# New approach to the retrieval of AOD and its uncertainty from MISR observations over dark water

Marcin L. Witek[1], Michael J. Garay[1], David J. Diner[1], Michael A. Bull[1], and Felix C. Seidel[1]

[1]Jet Propulsion Laboratory, California Institute of Technology, 4800 Oak Grove Drive, Pasadena, CA 91109, USA

Correspondence to: Marcin L. Witek (marcin.l.witek@jpl.nasa.gov)

## Abstract

A new method for retrieving aerosol optical depth (AOD) and its uncertainty from Multi-angle Imaging SpectroRadiometer (MISR) observations over dark water is outlined. MISR's aerosol retrieval algorithm calculates cost functions between observed and pre-simulated radiances for a range of AODs (from 0.0 to 3.0) and a prescribed set of aerosol mixtures. The previous Version 22 (V22) operational algorithm considered only the AOD that minimized the cost function for each aerosol mixture, then used a combination of these values to compute the final, "best estimate" AOD and associated uncertainty. The new approach considers the entire range of cost functions associated with each aerosol mixture. The uncertainty of the reported AOD depends on a combination of a) the absolute values of the cost functions for each aerosol mixture, b) the widths of the cost function distributions as a function of AOD, and c) the spread of the cost function distributions among the ensemble of mixtures. A key benefit of the new approach is that, unlike the V22 algorithm, it does not rely on empirical thresholds imposed on the cost function to determine the success or failure of a particular mixture. Furthermore, a new Aerosol Retrieval Confidence Index (ARCI) is established that can be used to screen high-AOD retrieval blunders caused by cloud contamination or other factors. Requiring ARCI≥0.15 as a condition for retrieval success is supported through statistical analysis and outperforms the thresholds used in the V22 algorithm. The described changes to the MISR dark water algorithm will become operational in the new MISR aerosol product (V23), planned for release in 2017.

## 1. Introduction

Uncertainty estimation in satellite remote sensing is a highly challenging endeavor that requires diligent assessment of many potential sources of error. Some of these errors are random and well understood and generally follow a traditional error propagation pathway. Other errors—which in underconstrained retrievals of geophysical quantities can be the dominant sources of uncertainty—are not readily assessed as they stem from various approximations and assumptions that lead to non-linear systematic responses. Additionally, there are sources of error that cannot be quantified at all that are attributable to the resolution of an instrument and variability at scales smaller than those observed.

Recently, Povey and Grainger (2015) provided a detailed overview and classification of possible sources of error in satellite retrievals. They clearly distinguish between pixel-level uncertainties associated with individual retrievals and bulk uncertainty metrics originating from validation studies. The two terms, however, are often not properly differentiated in the literature.

This is in part due to a lack of awareness about the issue as most studies focus on external validation of satellite data products, which often do not contain information about the uncertainties associated with individual retrievals. This status quo started changing due to growing pressure from the data assimilation community, which requires information on pixel-level uncertainties in order to make proper use of satellite-derived geophysical quantities within

the framework of a numerical model.

An example of a remotely sensed geophysical parameter that is being increasingly used in data assimilation applications is the aerosol optical depth (AOD)(Lynch et al., 2016; Shi et al., 2011, 2013; Zhang et al., 2008; Zhang and Reid, 2006, 2009, 2010). AOD represents the total extinction by aerosol particles in the atmospheric column from the surface to the top of the

atmosphere. Model data assimilation generally requires a value of AOD and an uncertainty associated with this value in order to determine whether or by how much the model should be adjusted to agree with the retrieval. AODs are readily available in many satellite data products, such as those from the Moderate Resolution Imaging Spectroradiometer (MODIS), the Multi-angle Imaging SpectroRadiometer (MISR), and the Visible Infrared Imaging Radiometer Suite (VIIRS),

whereas individual retrieval uncertainties, in many cases, are not. Pixel-level information often consists of retrieval quality assurance flags, which categorize the "usability" of a retrieval based on a qualitative judgment of the algorithm performance. For example, MODIS Collection 6 uses four coarse labels to communicate retrieval quality, ranging from 0 to 3, with 0 being the least trusted and 3 indicating the highest quality (Hubanks, 2015; Levy et al., 2013; Remer et al., 2005).

Other metrics conveying some useful information about AOD retrieval uncertainty are the proximity of clouds and cloud coverage in the retrieval region (Shi et al., 2011, 2013, 2014; Witek et al., 2013; Zhang and Reid, 2006). While such metrics are very valuable, they comprise only crude proxies for pixel-level uncertainties and, therefore, have limited quantitative utility in applications such as aerosol forecasting and data assimilation.

The most frequently used metric that quantitatively characterizes the quality of a particular AOD dataset as a whole is the error envelope (EE)(Bréon et al., 2011; Garay et al., 2016; Kahn et al., 2010; Levy et al., 2010, 2013; Limbacher and Kahn, 2014; Omar et al., 2013; Remer et al., 2008; Sayer et al., 2012, 2013; Witek et al., 2013). EE results from a validation study where a particular satellite AOD dataset is compared against another AOD dataset, typically ground-based

information from the Aerosol Robotic Network (AERONET) (Dubovik et al., 2000; Holben et al., 1998), which is considered to represent the "truth". Taking the general form of ±(a+b×AOD) (or max[±$a$, ±($b$×AOD)]), where $a$ and $b$ are empirically determined constants, EE represents the range required so that about 68% (one standard deviation) of the matched data agree, providing an overall characterization of the entire dataset. The EE is often regarded as an expected retrieval

error, or confidence envelope. External validation is fundamental to assessing a dataset's overall quality and serves as a useful guide for identifying the presence of systematic errors in the dataset

(e.g., Kahn et al., 2010). The EE, which describes the performance of the dataset as a whole relative to a reference dataset, however, does not represent and should not be confused with the uncertainty characteristics of an individual retrieval.

More appropriate attempts to address the true AOD retrieval uncertainty involved sensitivity tests of the retrieval algorithm with respect to varying external factors such as lower boundary conditions or aerosol microphysical properties (Kahn et al., 2001; Kalashnikova and Kahn, 2006; Sayer et al., 2016). These studies were limited in scope as they were unable to address all possible sources of error. Povey and Grainger (2015) suggested the use of ensemble techniques as a comprehensive means of quantifying uncertainties in satellite remote sensing of the environment. Each member of the ensemble adds a random perturbation to the measurements, ancillary parameters, and underlying retrieval assumptions in order to comprehensively map the probability distribution of the retrieved quantity. Even though such an approach would face significant conceptualization and computational challenges, especially in operational data processing, there are already examples of ensembles being used in aerosol remote sensing. One example is MISR (Diner et al., 1998), which employs many different particle microphysical mixtures as part of its operational aerosol retrieval process (Kahn et al., 2001; Martonchik et al., 1998). The resulting spread of AOD solutions for this ensemble of aerosol mixtures offers quantitative insight into the uncertainty of the individual retrieval. Such an approach, if extended to all poorly quantifiable nonlinear sources of error and physically plausible realizations of parameter space, has the potential of providing a robust and comprehensive measure of retrieval uncertainty in the manner suggested by Povey and Grainger (2015).

This study describes a new approach to determining AODs and AOD uncertainties in MISR retrievals. MISR is an instrument aboard the National Aeronautics and Space Administrations (NASA) Earth Observing System (EOS) Terra satellite that performs radiometric observations of the Earth using nine pushbroom cameras pointing at nine different angles (Diner et al., 1998, 2005). All nine cameras observe the same area on Earth within seven minutes, at four different wavelengths. The multi-angle viewing capability allows MISR to sample portions of the scattering phase function simultaneously, providing information that helps distinguish between different aerosol types (Kahn et al., 2001; Kalashnikova and Kahn, 2006). The retrieval process is based on the minimization of a cost function evaluated between the instrument observations and pre-calculated radiometric look-up tables (LUTs) that are in turn based upon a prescribed set of aerosol mixtures. Version 22 (V22) of the algorithm, which has been in operational production since December 2007, considers 74 aerosol mixtures with different microphysical properties that are intended to represent typical atmospheric conditions found on Earth (Kahn et al., 2009, 2010). MISR has two aerosol processing pathways, one for dark water (oceans, seas, deep lakes) and the other for the land surface (Martonchik et al., 2002). The modifications described in this study currently apply only to the dark water algorithm (Kalashnikova et al., 2013; Witek et al., 2013).

The paper is organized as follows. Section 2 describes the V22 operational approach to determining AODs and their uncertainties in MISR aerosol retrievals over dark water. In section 3 critical modifications and changes, which form the basis of a new methodology, are presented.

Section 4 introduces an important new metric that is employed to assess the quality of retrievals. This criterion is used for distinguishing between "good" and "poor" retrievals. Section 5 offers statistical analysis of the new AOD retrieval uncertainty and comparisons against the V22 approach. Finally, a short summary of the study follows in section 6.

## 2. Previous MISR V22 dark water algorithm

A detailed description of the MISR retrieval strategy is described in the Level 2 Aerosol Retrieval Algorithm Theoretical Basis document (Diner et al., 2008). The MISR aerosol retrieval algorithm follows two separate lines of processing depending on the surface type: dark water and land. The two retrieval types are independent of each other and largely rely on different physical and empirical assumptions. Only the dark water algorithm (Kalashnikova et al., 2013) is considered in this study. Here some key elements of the V22 algorithm relevant to the new approach are reviewed.

The problem of retrieving aerosol properties over large water bodies, such as oceans, seas, or deep lakes, is greatly simplified by the fact that reflectance from such surfaces is uniform and that such deep-water bodies are essentially black at red and near-infrared (NIR) wavelengths. One-dimensional radiative transfer (RT) theory is sufficient for determining the relationship between top-of-atmosphere (TOA) radiances and AOD. However, this calculation assumes an aerosol model that specifies the particle size distribution, shape, complex refractive index, and vertical distribution. Additional assumptions are made about the gaseous concentration in the atmosphere (ozone absorption, Rayleigh scattering) and surface whitecap fraction (i.e., the area of the surface covered by white foam from breaking waves). MISR's ability to observe multi-angle radiances, which are in large part governed by the shape of the aerosol scattering phase functions, provides a wealth of information with which to refine aerosol retrievals over dark water (Kalashnikova and Kahn, 2006).

The MISR aerosol retrieval algorithm relies on a LUT generated for a predefined set of mixtures with known optical properties. The V22 operational algorithm defines 74 aerosol mixtures, each of which consists of up to three unique particle types having prescribed optical and microphysical properties (Kahn et al., 2010; Kahn and Gaitley, 2015). The 74 mixtures consist of combinations of eight primary particle types. The MISR particle types and mixtures are designed to represent several compositional categories typically found in the atmosphere, such as sea spray, sulfate/nitrate, mineral dust, carbonaceous, and urban soot aerosols. Recently, Kahn and Gaitley (2015) provided a thorough verification of MISR's aerosol type retrieval capabilities and Lee et al. (2016) demonstrated that the MISR aerosol particle climatologies regionally showed good agreement with the results of chemical transport models. The current 74 mixture set, however, is not complete and has some documented deficiencies (Kahn et al., 2010; Kalashnikova et al., 2013; Limbacher and Kahn, 2014). A more comprehensive set of aerosol types and mixtures will be considered in a future release of the MISR's aerosol algorithm.

For each of the 74 mixtures, forward RT calculations are performed to provide top-of-atmosphere (TOA) radiances for the 36 MISR channels (9 angles and 4 wavelengths). These

radiances are stored in the form of "equivalent reflectances" in the Simulated MISR Ancillary Radiative Transfer (SMART) Dataset, where equivalent reflectance is defined as

$$\rho = \frac{\pi L}{E_0},$$ 

(1)

where $L$ is the radiance and $E_0$ is the exo-atmospheric solar irradiance determined for each MISR spectral band. The RT calculations of TOA radiances are carried out for discrete values of mixture optical depth, from 0 to 3, referenced to MISR's 558 nm (green) band for all plausible combinations of view and solar geometries. The simulations incorporate a modified linear mixing theory for mixtures containing more than one aerosol type, a wind-speed-driven glitter and whitecap model, ozone correction, and Rayleigh scattering (Abdou et al., 1997). The modeled TOA radiances are then directly compared with the MISR observations. The criterion used to find the best-fitting optical depth for a particular mixture is minimization of the reduced $\chi^2_{abs}$ parameter, calculated as a function of green-band AOD ($\tau$)

$$\chi^2_{abs}(\tau) = \frac{\sum\limits_{l=1}^{4} w_l \cdot \left[ \sum\limits_{j=1}^{9} v(l,j) \cdot \frac{\left[ \rho_{MISR}(l,j) - \rho_m(l,j) \right]^2}{\sigma^2_{abs}(l,j)} \right]}{\sum\limits_{l=1}^{4} w_l \cdot \left[ \sum\limits_{j=1}^{9} v(l,j) \right]}.$$

(2)

In Eq. 2, $\rho_{MISR}$ are MISR equivalent reflectances, $\rho_m$ are modeled TOA equivalent reflectances for a given aerosol mixture, and $\sigma_{abs}$ are the absolute radiometric uncertainties in $\rho_{MISR}$ calculated as $\sigma_{abs}(l,j) = 0.05 \max\left( \rho_{MISR}(l,j), 0.04 \right)$. The summation index $l$ is over the 4 MISR wavelengths and $j$ is over the 9 MISR cameras. The parameter $v(l,j) = 1$ if a valid value of $\rho_{MISR}(l,j)$ exists, and is set to 0 otherwise. The weights $w_l$ are always equal 1 for the red and NIR bands, and are $\leq 1$ for the blue and green bands, depending on $\tau$. These weights allow individual bands to contribute varying amounts to the $\chi^2_{abs}$ calculation as a function of optical depth. For example, at low $\tau$ (<0.5) the MISR 446 nm (blue) and 558 nm (green) bands are not used in $\chi^2_{abs}$ calculations since at these wavelengths and $\tau$ values the water leaving radiance could be a significant contributor to the TOA signals.

For each mixture, the best fitting value of $\tau_{mix}$ is taken to be the value that minimizes $\chi^2_{abs}$ using a parabolic fitting approach (Diner et al., 2008). Additional parameters are used to determine the goodness of fit of the particular aerosol mixture to the MISR data. Those are $\chi^2_{geom}$, $\chi^2_{spec}$, and $\chi^2_{\max dev}$ which are calculated at the previously obtained value of $\tau_{mix}$. Definitions of these parameters can be found in Diner et al. (2008) and Kalashnikova et al. (2013). An aerosol mixture is considered "successful" if all four metrics $\chi^2_{abs}$, $\chi^2_{geom}$, $\chi^2_{spec}$, and $\chi^2_{\max dev}$ do not exceed certain empirically established thresholds. In V22, these thresholds are set to 2, 3, 3, and 5, respectively.

As a result of this procedure, each retrieval region is assigned a set of mixtures and associated AODs that pass all the threshold criteria. In the special case when none of the 74 mixtures pass the threshold tests, the retrieval is considered unsuccessful and no AOD is reported. In most cases, however, multiple mixtures are deemed successful. They typically have

somewhat different $\tau_{mix}$ values corresponding to their minimum $\chi^2_{abs}$. The arithmetic mean of all AODs from the passing mixtures is reported as the "best estimate" AOD in the V22 Level 2 aerosol product with the field name "RegBestEstimateSpectralOptDepth". The retrieval uncertainty, with the field name "RegBestEstimateSpectralOptDepthUnc", is determined from the standard deviation of the AODs from the passing mixtures. In the case where only a single mixture is

successful, the uncertainty is determined directly from the parabolic fit for that mixture (Diner et al., 2008).

A critical aspect of the retrieval process and its outcome is its dependence on a number of empirically determined thresholds. The specific numerical values for the $\chi^2_{abs}$, $\chi^2_{geom}$, $\chi^2_{spec}$, and

$\chi^2_{\max dev}$ thresholds were chosen based on pre-production tests aimed at eliminating obvious

blunders while maintaining adequate spatial coverage. However, the reported AOD and its uncertainty are directly linked to these thresholds, which entails a certain degree of subjectivity. For example, a mixture and AOD combination resulting in a $\chi^2_{abs}$ value of 1.99 would be

considered successful, while a mixture/AOD combination with a $\chi^2_{abs}$ value of 2.01 would not be. Alternatively, mixtures with very different AODs, for example a non-absorbing and an absorbing

mixture, might both be considered successful and be included in the uncertainty calculation, but have dramatically different $\chi^2_{abs}$ values, something which is not taken into account when

determining the uncertainty. Mitigating such issues was an important driver for developing a new approach to AOD determination and its uncertainty for MISR dark water retrievals in a more objective manner.


## 3. New approach to AOD retrieval and its uncertainty

The empirical thresholds in goodness-of-fit parameters in the V22 MISR dark water aerosol retrieval algorithm are used to select successful aerosol mixtures. This affects the frequency of

retrieval success as well as the resulting AODs, AOD uncertainties, and aerosol properties. A more desirable approach would minimize the reliance on empirical thresholds. In this section, a new method is described that meets this objective and simplifies the retrieval of the "best estimate" AOD and its associated uncertainty. Furthermore, it results in a single parameter that enables screening of retrieval blunders and AOD outliers and which outperforms results derived using the

original V22 thresholds. This algorithm revision has been implemented in the software used to generate the next version of the MISR operational aerosol product, V23, scheduled for public release in 2017.

The new method relies solely on the $\chi^2_{abs}$ values calculated using Eq. 2. The other goodness-of-fit metrics are retained solely for diagnostic purposes. Extensive testing showed that AOD

selection in V22 was governed primarily by $\chi^2_{abs}$, with the other parameters typically having little effect due to the magnitude of their thresholds, except in a limited set of cases. The key elements of the new method are visualized in Figure 1 using actual MISR data from a randomly selected case. First, the values of $\chi^2_{abs}$ for each mixture are calculated as continuous functions of $\tau$. The

result is then inverted, yielding the distribution of $1/\chi^2_{abs}$. Taking the reciprocal has two primary benefits. First, it gives a smaller weight to retrievals with large values of $\chi^2_{abs}$ that represent poor agreement between the model and the MISR observations. Second, the distribution of $1/\chi^2_{abs}$ tends to look Gaussian, with a peak at $\tau_{mix}$. In the next step, these functions are averaged over all $N=74$ mixtures, leading to:

$$f(\tau) = \frac{1}{N}\sum_{m=1}^{N}\frac{1}{\chi^2_{abs,m}(\tau)} \,. \qquad\qquad (3)$$

The position of the peak of the average distribution $f$ is the new retrieved AOD

$$AOD = \tau\big(\max\big(f(\tau)\big)\big). \qquad\qquad (4)$$

The function $f$ can be interpreted as a probability density function (PDF) for AOD. The most likely AOD is the one that maximizes $f$ (Eq. 4), and the retrieval uncertainty is related to the width of the

PDF. The function $f$, which in most cases closely resembles a Gaussian (normal) distribution, has a peak that is narrow or wide depending on how closely the individual $\tau_{mix}$ from the 74 mixtures are clustered. Furthermore, because the absolute values of $1/\chi^2_{abs,m}$ from each mixture contribute to the overall shape of the PDF, aerosol models fitting the observations well (low $\chi^2_{abs}$) dominate the shape of $f$ and the position of its peak, whereas mixtures with poor fits (high $\chi^2_{abs}$) contribute less.

The retrieval uncertainty ($\sigma$) is determined from the full width at half maximum (FWHM) of $f$, and scaled to a standard deviation under the assumption that the functional form of $f$ in the vicinity of its peak can be approximated by a Gaussian distribution:

$$\sigma = \frac{FWHM(f)}{2\sqrt{2\ln 2}} \approx \frac{FWHM(f)}{2.3548} \,. \qquad\qquad (5)$$

      Equations 4 and 5 form the backbone of the new approach for determining AODs and their

uncertainties in MISR retrievals over dark water in the V23 algorithm. One important benefit of the method is that it does not rely on empirically determined thresholds. In all cases, all 74 mixtures contribute to the retrieved AOD, but the amount they contribute depends on how well they agree with the MISR observations. The retrieval uncertainty is then related to the degree to which the AODs associated with the entire ensemble of aerosol mixtures cluster around a specific

AOD. If all mixtures are consistent with the same AOD and are highly sensitive to its specific value, the peak in $f$ will be narrow and the uncertainty low. If mixtures disagree as to a single value of AOD, or the $\chi^2_{abs}$ parameter is relatively insensitive to the AOD, the distribution will be broad and the reported uncertainty will be larger.

While the width of the average distribution $f(\tau)$ contains information about the retrieval uncertainty, the peak of the distribution, $\max(f(\tau))$, has an additional important benefit: it can be utilized as a retrieval screening parameter. $\max(f(\tau))$ represents the overall agreement of the TOA equivalent reflectances from the aerosol mixtures in the LUT with the MISR observations,

which can be considered a measure of the confidence in the retrieval. $\max(f(\tau))$ is designated the "Aerosol Retrieval Confidence Index", or ARCI. Low ARCI implies that generally high $\chi^2_{abs}$ were obtained, indicating that the aerosol models fit the MISR observations poorly. Large ARCI, on the other hand, means that for some models sufficiently low $\chi^2_{abs}$ were obtained, signifying good agreement with the observations. In a sense, a threshold on ARCI is similar to a threshold on $\chi^2_{abs}$,

except that the former incorporates all aerosol mixtures simultaneously while the latter is applied mixture by mixture. Furthermore, as will be shown in section 4, ARCI is more effective than $\chi^2_{abs}$ in filtering out retrieval blunders and other obvious outliers.

Figure 1 visualizes the important steps of the method using actual MISR data. In this example, the new retrieved AOD is 0.182, whereas the V22 method gave a value of 0.174. The new

retrieval uncertainty is 0.049, which is more realistic than the 0.003 uncertainty reported by the V22 algorithm. The very small uncertainty in V22 is due to the fact that only two mixtures were considered successful by passing the V22 thresholds. This example highlights a deficiency in the V22 assessment of retrieval uncertainty as the uncertainty is highly dependent on the number of passing mixtures as well as the value of the four separate thresholds used to determine which

mixtures are considered successful. The new procedure eliminates the need for thresholds in determining AOD and its uncertainty, and the only threshold involved is applied to the single ARCI parameter, which is used as a retrieval quality indicator.

**4. Retrieval quality assessment**

In the MISR V22 retrieval algorithm several thresholds were set to filter out mixtures that do not provide a good match to the instrument observations. The threshold that provides the most strict screening in V22 is $\chi^2_{abs} \leq 2$, which is applied individually to each aerosol mixture. Because the thresholds provide an additional line of defense against clouds that were not screened by other

procedures in the aerosol retrieval process, elimination of these thresholds can result in a large number of high-AOD retrievals in areas that are notorious for frequent cloud cover, but have climatologically very low AODs.

This situation is illustrated in Figure 2, which shows the average AOD for the combination of January and July of 2007 obtained with the new retrieval methodology. Vast areas of the high-

latitude oceans are speckled with unrealistically high AODs, clearly indicating an issue with cloud contamination. In V22 the $\chi^2_{abs}$ and other thresholds are able to limit the occurrence of such blunders. In the new algorithm, which performs aerosol retrievals on a 4.4 km grid, in contrast to

the coarser 17.6 m grid used in V22, the problem of cloud contamination is further amplified due to closer proximity to cloud edges. Applying the same thresholds as in V22 does not fully mitigate the issue: substantially more 4.4 km retrievals remain cloud contaminated than in V22 (results not shown). Fortunately, the ARCI metric introduced in the previous section proves to be
extremely effective at filtering out potentially cloud-contaminated AOD retrievals.

        Figure 3a shows average AOD from 4.4 km retrievals as a function of the minimum value of $\chi^2_{abs}$. In total about 49 million retrievals were analyzed here. After a rapid initial drop related to a similar rapid increase in sampling (Fig. 3b), the average AOD increases gradually with increasing $\min\left(\chi^2_{abs}\right)$, while the sampling continues decreasing. The AOD increase could be due to a
combination of the increasing number, magnitude, and relative occurrence of cloud-contaminated, high-AOD retrievals with increasing $\min\left(\chi^2_{abs}\right)$. Based on the V22 $\chi^2_{abs}$ threshold approach, all retrievals with $\min\left(\chi^2_{abs}\right) \le 2.0$ would have been considered successful. However, no clear justification for a threshold, either at 2.0, or any other value, is evident in the average AOD data. Choosing a value for the threshold that minimizes the average AOD would screen clouds but
also potentially screen optically thick aerosol plumes, such as the heavy dust that is prevalent off the west coast of Africa. The picture looks different, however, when one considers the average AOD as a function of ARCI (Fig. 3c). After excluding the initial fluctuation for extremely small ARCI related to poor sampling, two distinct regimes in the trend of average AOD can be noticed. In the first regime, the average AOD is highly sensitive to the specific value of ARCI, characterized by a
sharp decrease in AOD with increasing ARCI between about 0.03 and 0.13. This suggests that a decreasing number of cloud-contaminated, high-AOD retrievals are included in the average as the ARCI is increased. Indeed, the percentage of retrievals with AOD higher than 2.0 reaches its peak, 16%, at ARCI equal to 0.03, and decreases to about 2% when ARCI is 0.13. In the second regime, there are relatively small changes in the average AOD as ARCI increases above 0.15. The low AOD
gradient in the second regime suggests a low prevalence of cloud contaminated or erroneous AODs. The retrieval count decreases slowly with increasing ARCI (Fig. 3d), indicating that the observed trends in the average AOD cannot be ascribed to a change in frequency. Conveniently, the number of screened retrievals with ARCI ≥ 0.15 is similar to the number of retrievals that do not pass the $\chi^2_{abs} \le 2.0$ threshold. Out of about 49 million retrievals, 35.9% are below the ARCI
threshold (not passing), and 37.1% are above the $\chi^2_{abs} \le 2.0$ threshold. We set ARCI ≥ 0.15 as the value to be used as a threshold for screening retrieval blunders due to potential cloud contamination or other factors.

        Another way to look at the difference between the two screening approaches is presented in Fig. 4a, which shows the two-dimensional distribution of average AOD as a function of $\min(\chi^2_{abs}$
) and ARCI using combined data from January and July of 2007. Figure 4b shows the respective retrieval count. The previous $\chi^2_{abs}$ threshold limit at 2.0 is marked with the black vertical line. All retrievals to the left of this line would have been considered successful in the V22 algorithm. This

includes a small group of high-AOD retrievals with min($\chi^2_{abs}$) close to 0.2 and ARCI about 0.1. Another suspicious group of retrievals with high average AODs that would have passed the previous threshold is close to min($\chi^2_{abs}$) = 2.0. The new ARCI threshold limit, marked with the gray horizontal line, eliminates most of the suspiciously high-AOD regions. All retrievals above the

gray horizontal line are considered to be of sufficiently good quality. Of course, more complicated relationships could be investigated, but the use of ARCI as a single screening parameter proves to be highly efficient and furthermore has the advantage of simplicity. The V23 MISR aerosol product will provide the values of both ARCI and $\chi^2_{abs}$ for use in exploring custom-made cloud screenings and for other purposes.

10       Figure 5 presents the average AOD distribution obtained using the combination of the January and July 2007 data with retrieval screening based on the ARCI metric (ARCI≥0.15). This result is directly comparable to Fig. 2, which uses the unscreened data. The benefit of ARCI screening is readily apparent. AODs in large swaths of remote oceans are now represented by smaller and more realistic values (Witek et al., 2013). At the same time, climatologically large

AODs off the coasts of Africa and South and East Asia are retained, indicating that the new screening method does not unintentionally remove all high AODs that are likely valid. The global average AOD is reduced from 0.295 for the unscreened data to 0.141 with ARCI screening. However, speckles of high AOD values are still present in many remote and cloudy regions. The majority of these retrievals are visibly cloud contaminated. This demonstrates that the ARCI

screening is not ideal as some erroneous AODs pass the threshold. Increasing the threshold reduces the appearance of blunders, but also decreases the number of valid low- and moderate-AOD retrievals, reducing the overall coverage. Because the choice of setting the ARCI threshold limit at 0.15 is well supported statistically (see Figs. 3c, 4a), the remaining cloud-contaminated AOD retrievals should be addressed using another screening method. A possible approach is to

employ the clear flag fraction metric discussed in Witek et al. (2013). The application of this approach to removing the remaining cloud-contaminated retrievals in the MISR V23 aerosol product will be discussed in a separate paper.

**5. Statistical assessment of AOD retrieval uncertainty.**
The AOD retrieval uncertainty described by Eq. 4 is a measure of the sensitivity of the algorithm to the assumed aerosol microphysical properties. This is an important factor affecting retrieval uncertainty (Li et al., 2009; Povey and Grainger, 2015), but, as mentioned in the introduction, there are many other sources of error not accounted for in this approach. Hence, the AOD

uncertainty obtained from the algorithm should not be interpreted as a measure of how far the retrieved AOD deviates from the "truth". This is an important distinction that needs to be properly understood. The calculated uncertainty is purely algorithmic and depends on the initial choice of aerosol mixtures that go into the MISR SMART LUT. Any changes in the prescribed mixtures would lead to different uncertainty estimates. Furthermore, if Eq. 2 is modified such that a

different goodness-of-fit metric is used, a different uncertainty result would be expected. For

these reasons, interpretation of the established uncertainty does not extend beyond the algorithm's performance. It does, however, help establish confidence intervals on the retrievals when comparing one pixel to another.

In Figure 6, AOD uncertainty is plotted as a function of AOD using the combined data from January and July of 2007. Only ARCI-screened retrievals are considered. Reported uncertainties are generally much smaller than their associated AODs. Only for very low AOD values do the uncertainties exceed the retrieved AODs. The linear fit to the data indicates that the uncertainty is about 12% of the AOD and has an offset of 0.012. This offset is much smaller than what has been traditionally discussed in the literature, specifically the 0.03 or 0.05 in the smaller and larger EE, respectively (e.g. Kahn et al., 2010). At low AODs (<0.03) the reported uncertainty is almost always below these offset levels. At higher AODs, uncertainties are often smaller than 20% or even 10% of the retrieved AODs. Note that there is always a substantial spread of AOD uncertainties at any given AOD level, often over an order of magnitude. This shows that, at least from a statistical perspective, the algorithm is capable of representing variability in retrieval confidence. For example, a retrieved AOD of 0.1 can have an uncertainty of 0.05 or of 0.005 depending on circumstances. Assigning physical meaning to a particular uncertainty value as a departure from the true value, as stated earlier, is a task that needs to be addressed separately.

A comparison of the new AOD uncertainties against their V22 predecessors reveals many similarities between the two, as evidenced in Figure 7. Recall that the V22 algorithm calculated uncertainties based on the AOD that minimized $\chi^2_{abs}$ for each mixture, while the V23 algorithm evaluates the full range of $\chi^2_{abs}$ as a function of AOD, so this agreement is not accidental. On average the V23 uncertainty is larger than that reported in V22. There does not seem to be a lower limit on uncertainties in V22, often exhibiting values below $10^{-3}$ whereas in V23 they mostly stay above $2\times10^{-3}$. The small values reported in V22 may be due to situations where only a single mixture was considered successful. Furthermore, there is discernable quantization, or clustering, of uncertainties in V22, visible as vertical striping in Fig. 7. This quantization is clearly eliminated in V23. Overall, the new AOD uncertainties appear to have a more reasonable statistical behavior compared to the uncertainties obtained in V22.

## 6. Conclusions and summary.
Ensemble techniques have been widely used in weather forecasting applications and climate research. They are indispensable in characterizing uncertainties and errors of highly non-linear systems, where standard error propagation techniques cannot be applied. These techniques are also useful tools for quantifying uncertainties in satellite remote retrievals of geophysical quantities (Povey and Grainger, 2015). MISR's aerosol retrieval strategy is a good example of the application of ensemble techniques to retrieval uncertainty assessment in operational data processing.

MISR's aerosol retrieval algorithm uses minimization of a cost function between observations and pre-calculated signals as a function of AOD. The spread of the cost function

around a particular AOD value is one indication of the uncertainty of the retrieved solution. Additionally, an ensemble of cost functions for different aerosol mixtures samples sensitivity of the retrieval process to the assumed aerosol optical and microphysical properties. This is one of the major sources of uncertainty in passive remote sensing of AOD. By including an ensemble of
aerosol types in the retrieval approach, an algorithmic measure of AOD retrieval uncertainty that includes the impacts of measurement uncertainties, model errors, and aerosol type variability can be effectively derived using MISR data.

This study presents a new approach to determining AODs and AOD uncertainties in MISR retrievals. The new method will become operational for dark water aerosol processing in the
upcoming release of V23 of the MISR aerosol product (scheduled for 2017). Unlike the V22 algorithm, the new approach eliminates several empirical thresholds. Instead, the AOD and AOD uncertainty determination relies solely on the $\chi^2_{abs}$ metric defined by Eq. 1. All considered mixtures contribute to the final result with a varying influence depending on the shape and magnitude of the associated cost functions. This approach allows for a consistent calculation of
AOD and AOD uncertainty without the need for screening acceptable mixture solutions based on a complex interplay of multiple, and somewhat arbitrary, thresholds.

An unintended side effect of the new retrieval approach is an increased abundance of (mostly) cloud-contaminated, high-AOD retrievals in oceanic areas where very low aerosol concentrations are expected. Those blunders—remnants of imperfect cloud screening—were also
present in V22, but many were rejected through the use of thresholds on different cost functions. They are more apparent in the V23 results due to the increase in the spatial resolution of the product from 17.6 km in V22 to 4.4 km in V23. Fortunately, an effective screening criterion has been established that filters out most cloud-contaminated retrievals. An analysis of the ARCI metric strongly suggests a specific threshold value, below which the retrievals become
increasingly contaminated by clouds. Although this screening method does not eliminate all AOD outliers, it is superior to the previously used thresholds in the V22 of the MISR aerosol product. Additional cloud screening making use of the clear fraction flag with retrieval regions is built into the V23 algorithm, and will be described separately. Comparison of the new V23 algorithm results (AODs and AOD uncertainties, in particular) to other products, specifically AERONET
sunphotometer measurements, will be addressed in future publications.

**Acknowledgement.**
The research was carried out at the Jet Propulsion Laboratory, California Institute of Technology,
under a contract with the National Aeronautics and Space Administration. Support from the MISR Project is acknowledged. We thank Ralph Kahn for providing helpful comments on the manuscript.

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

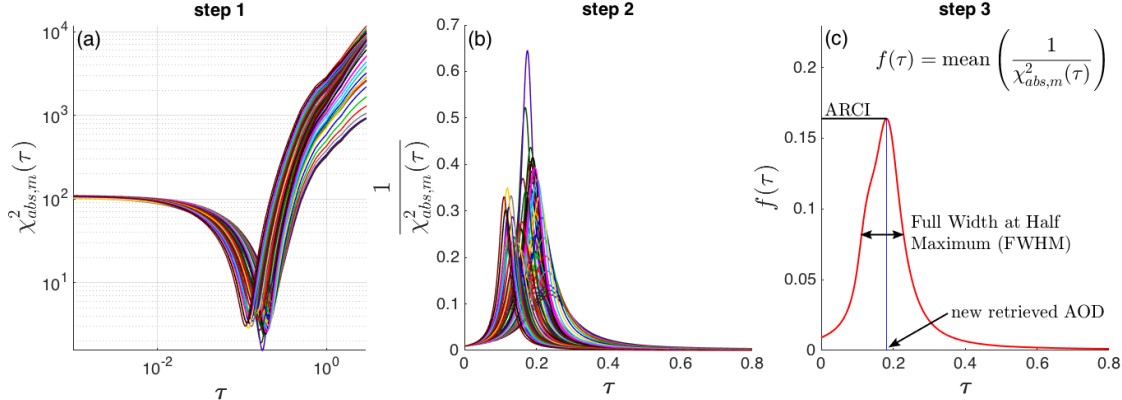

**Figure 1 Example of calculation steps performed in the new methodology for determining AOD and its uncertainty. (a)**
$\chi^2_{abs}$ **values for 74 MISR mixtures as a function of AOD ($\tau$) (Eq. 1); (b) inverse (reciprocal) values for the 74 mixtures; and (c) inverse residuals averaged over all mixtures (Eq. 2), with the new retrieved AOD, ARCI, and FWHM indicated on the distribution. The x-axis scale is logarithmic in panel (a) for a better visualization of the cost function at low $\tau$.**

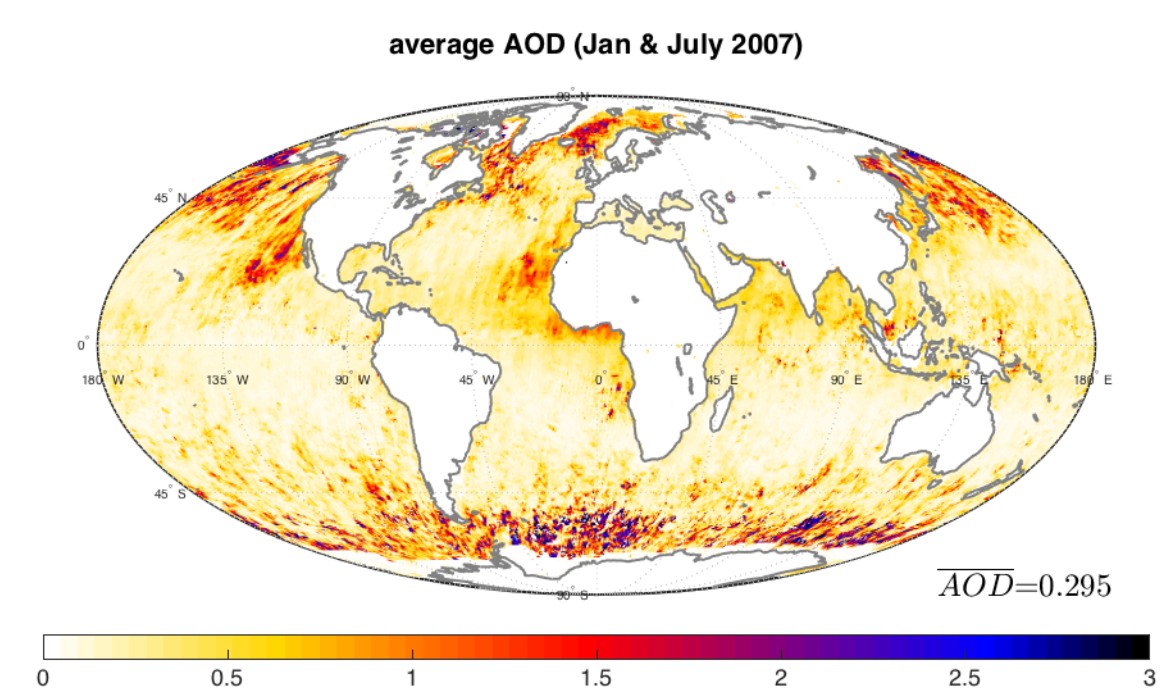

**Figure 2 Average AOD obtained using unscreened data from January and July of 2007. The data are mapped to a 0.5º × 0.5º grid. High AOD values over remote oceans indicate issues with cloud contamination in the retrieval process.**

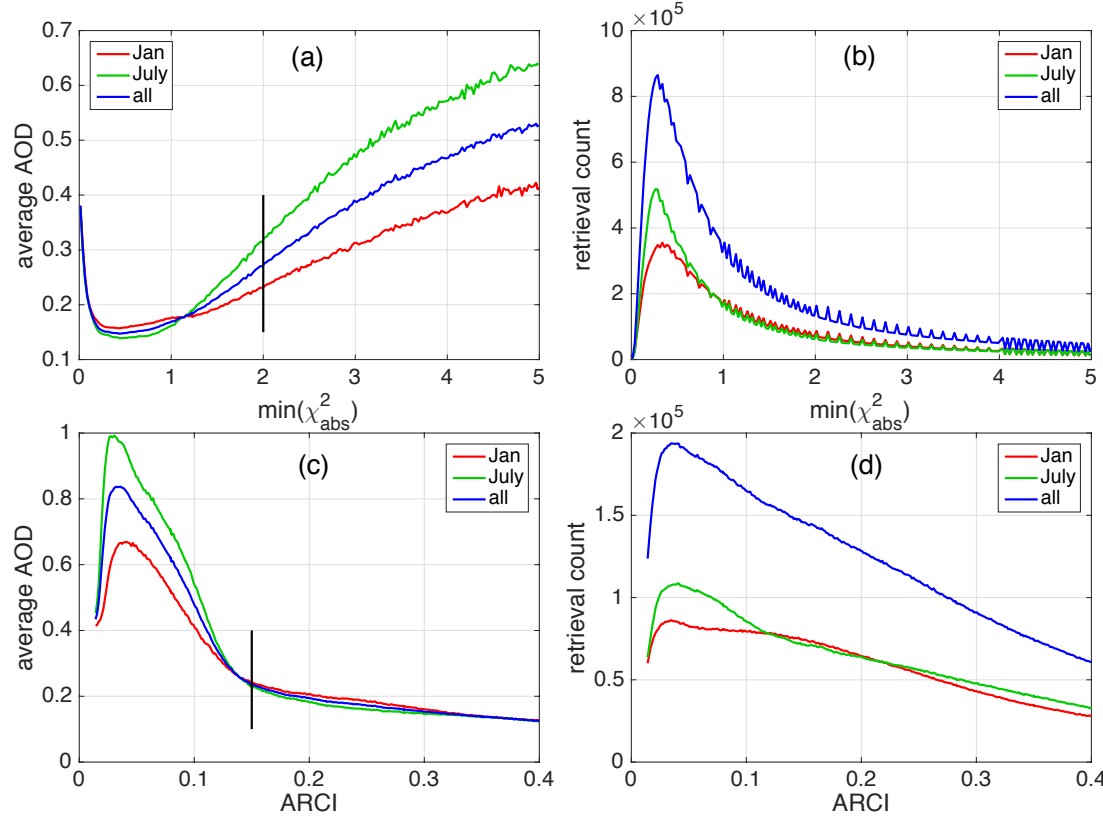

**Figure 3 Histograms of average AOD as a function of (a) min( $\chi^2_{abs}$ ), and (c) ARCI for January 2007, July 2007, and the two months combined. Panels (b) and (d) are histograms of retrieval counts corresponding to min( $\chi^2_{abs}$ ) and ARCI values, respectively.**

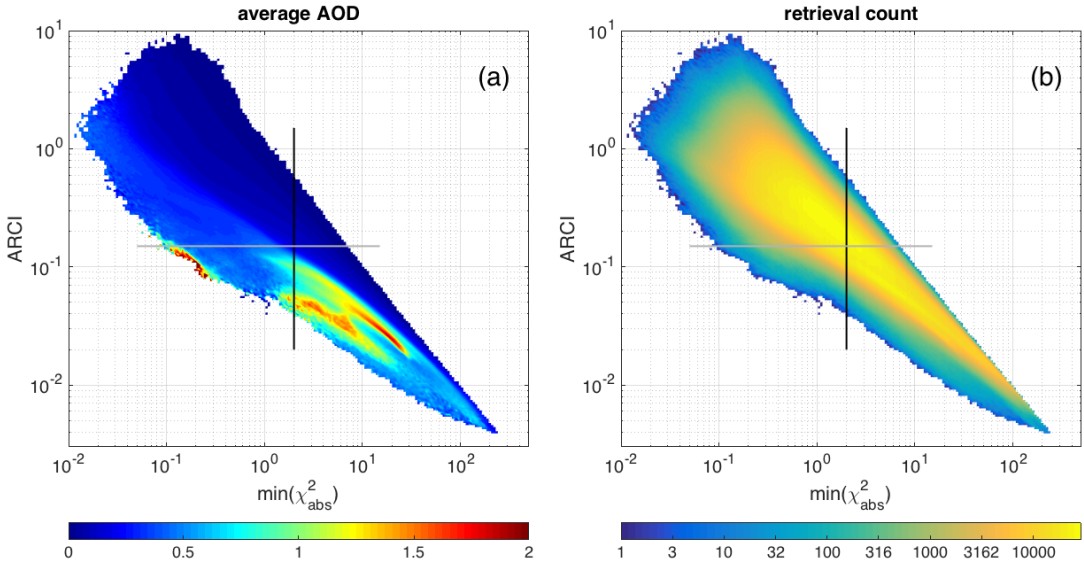

**Figure 4 (a) average AOD as a function of ARCI and min( $\chi^2_{abs}$ ) for the combined months of January and July of 2007, (b) retrieval count for the data plotted in panel (a).**

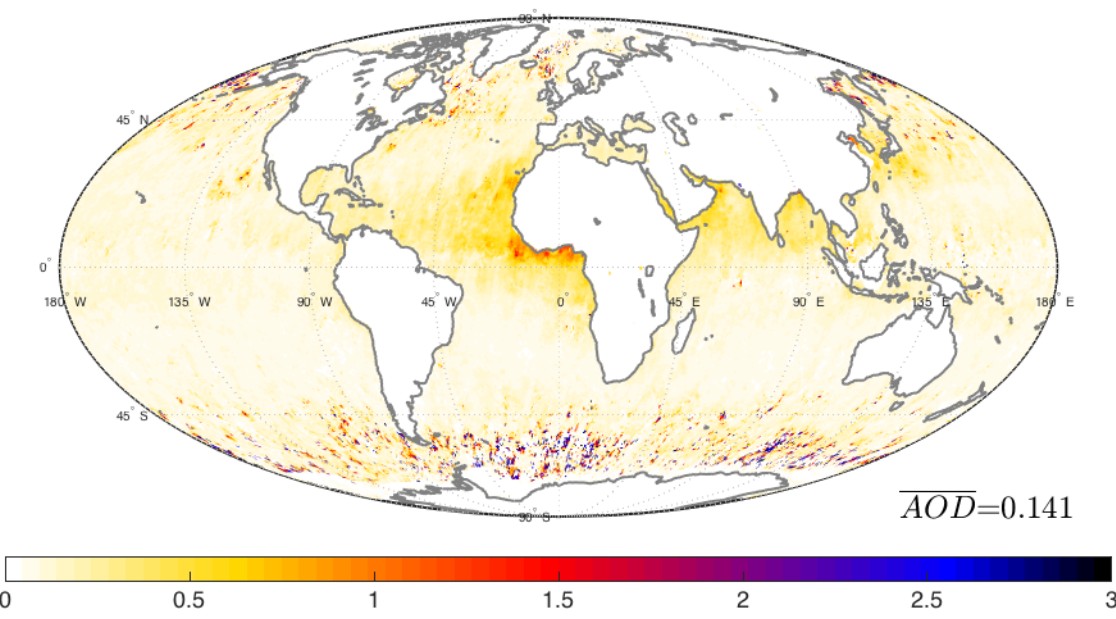

**average AOD with ARCI screening (Jan & July 2007)**

$\overline{AOD}=0.141$

**Figure 5 Average AOD distribution with ARCI≥0.15 screening for the combination of January and July 2007. The data are mapped to a 0.5º × 0.5º grid. There is substantial improvement in the global distribution of mean AODs when compared to the unscreened data in Fig. 2. However, some residual high AOD values remain over remote oceans. They can be further screened using other approaches.**

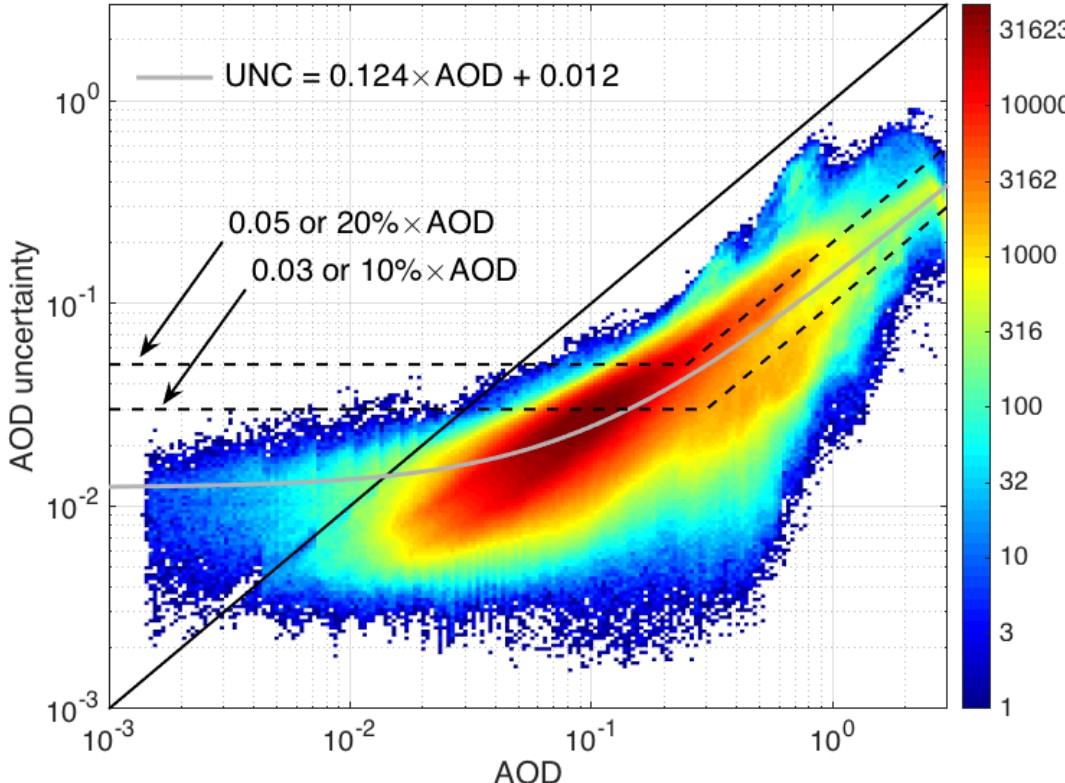

**Figure 6 Density plot of AOD uncertainty in log-log space as a function of AOD for the combined January and July 2007 data with ARCI screening. The black line is the 1-to-1 line, included as a visual guide to illustrate that, over most of the AOD range, the uncertainties are smaller than the AOD values themselves. The gray line is a linear fit to the data. Two dashed lines represent two arbitrary uncertainty envelopes.**

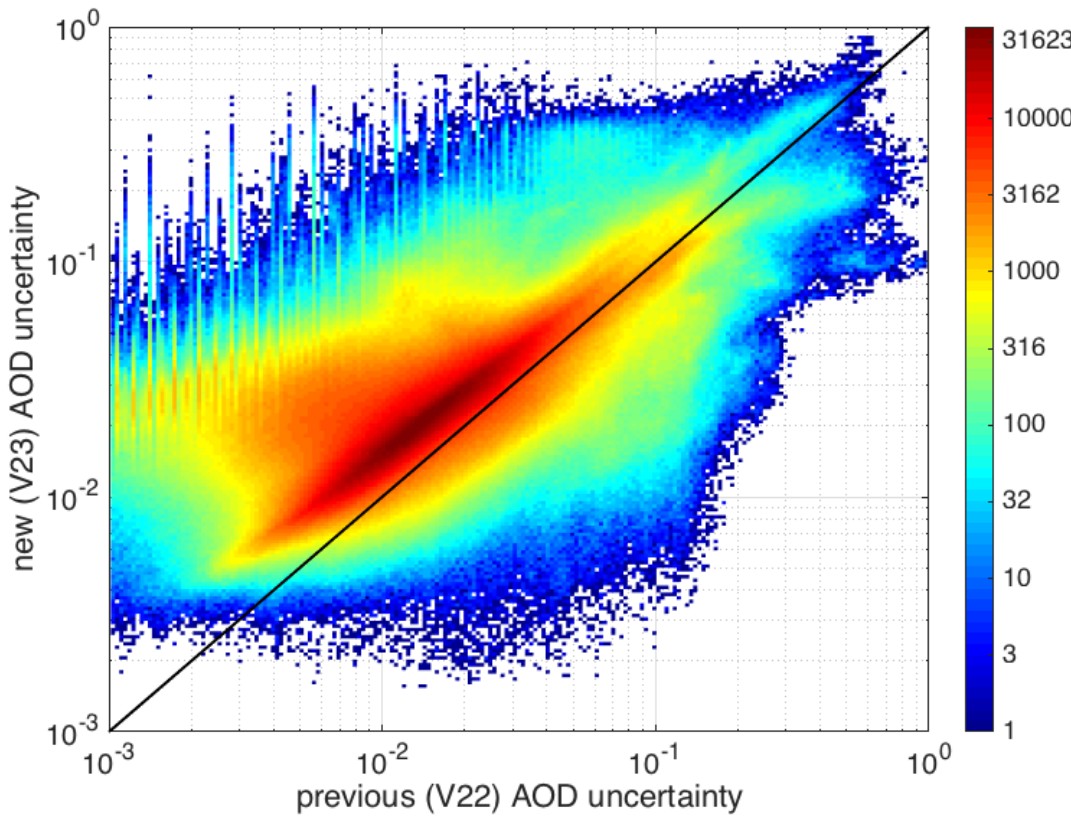

**Figure 7 Density plot showing comparison between the previous (V22) AOD uncertainty and the new (V23) AOD uncertainty. The black line is the 1-to-1 line.**