# Peer review of "New approach to the retrieval of AOD and its uncertainty from MISR observations over dark water"

_Atmospheric Measurement Techniques, 2017_

## Referee Comment (RC1) · Anonymous Referee #3 · 21 Sep 2017

This manuscript details a new method to retrieve AOD and pixel-level AOD uncertainty from the Multi-angle Imaging SpectroRadiometer. The manuscript is well-written, and it is likely that this new method will serve the MISR aerosol community well in the future. However, I can not support publication of this work without at least a cursory attempt at validation, which this paper is sorely missing.

Major Comment: Although I believe that this new method probably represents a substantial improvement for aerosol remote sensing from MISR, it is incumbent on the authors to prove this. The authors claim that not all sources of error are included, therefore no comparison of the AOD differences and pixel-level uncertainties (against

[Figure]

MAN or AERONET, for instance) should be done. If this new method is going to be implemented in the next version of the MISR aerosol retrieval algorithm (and going to be published here), it should be validated first. A comparison of new algorithm results with old algorithm results does not replace real validation.

Minor Comment:

The authors mention (5 times) that several of the thresholds from the current version of the MISR aerosol retrieval algorithm are arbitrary. Please refrain from so much hyperbole in the manuscript, as most empirical thresholds could be considered arbitrary (including your own ARCI threshold).
* * *

---

## Referee Comment (RC2) · Anonymous Referee #1 · 21 Sep 2017

This paper describes a change to the MISR aerosol retrieval algorithm. They select an ensemble of aerosol types and, for each, compute the radiances that would be observed at a range of aerosol optical depths (AOD). Previously, ensemble members were evaluated separately so each gave an AOD and cost, which were then filtered and averaged to calculate the final product. This paper proposes minimising a single cost function (being the sum of the individual cost functions) to find the AOD and it's uncertainty. The technique is rationalised based on two months of observations and is shown to produce more believable uncertainties, on average, than the previous algorithm.

[Figure]

I recommend this paper for publication after minor revisions. The technique proposed is definitely a step in the right direction and the paper is superbly drafted. However, the technique and description thereof could be improved by a more statistical approach. The paper justifies itself with qualitative descriptions of global averages and internal metrics rather than any validation activity, which is common but always disappointing. Specific comments on the paper are listed below, with some minor details collected at the end. The notation $PxLy$ refers to line $y$ of page $x$.

- My experience is in optimisation. One defines a cost function and selects an algorithm to efficiently search the 'surface' of that function for its global minima. The uncertainty is a measure of the 'width' of that minima in multi-dimensional space (i.e. the magnitude by which a variable could be changed without significantly increasing the cost). The cost function is usually the RMS difference between some modelled value and a measurement. If the model is accurate and the measurement suffers only random noise (of known variance), the minimal value of the cost function will sample a $\chi^2$ distribution, from which one can determine the probability that this measurement fit that model.

  To me, this paper essentially proposes that $f(\tau)$ is a probability density function (PDF) for AOD and that it is normally distributed. It follows that the most likely AOD is the $\tau$ that maximises $f$ and the uncertainty is the function's width. The proposed ARCI threshold can then be understood as eliminating retrievals that are exceedingly unlikely. Describing the problem with these basic statistical concepts could vastly simplify the paper, avoiding awkward phrasing like P8L5.

- Because this is a fairly straightforward statistical problem, there exists a variety of tools to check that (a) $f$ is in fact a good model of the PDF, (b) $f$ is normally distributed, and (c) the selected aerosol models are an unbiased sampling of the complete state space of real-world aerosols. A brief discussion of some of those points could provide a standardised means to evaluate your assumptions and

avoid qualitative judgements, such as the function 'closely resembles a Gaussian' (P7L12).

- Are you tabulating $f$ as a function of linear or log $\tau$? Figure 1 uses both as an $x$-axis, which is misleading. It should be logarithmic as AOD is log-normally distributed (which is clear from the asymmetry about $\tau_{max}$ in Fig.1(3)). If you're using linear space, you will underestimate the uncertainty and overestimate the mean.

- Why is there no validation of the new algorithm? It seems fairly substantial to move from averaging a few aerosol types per pixel to averaging 74. A few comparisons against AERONET or MODIS would be fine for a paper like this. A simple comparison of V22 vs. V23 would be a start, considering you did it for the uncertainty!

- In Sec. 3, you implicitly assume that the choice of aerosol type overwhelms any measurement error. Could Fig. 1 be adapted to show the sensitivity of a $\chi^2$ curve to typical measurement error? I'd expect it to move the curve slightly, but much less than the spread between curves.

P10L24 I'm unhappy with this paragraph.

L27 I think this is trying to distinguish between a validation activity, which you sadly aren't doing, and an uncertainty estimate, which you are. By definition, uncertainty is a parameter describing the range of values that can be reasonably ascribed to the quantity that is being measured. I believe that provides a 'measure of how far the retrieved AOD deviates from the "truth"'. The distinction is that uncertainty is a prediction of that difference while validation is a direct calculation of it.

L30 It's good to be clear that the estimated uncertainty is sensitive to the way you solve the problem. However, you don't tell the user what to do with that

information. I think a rational response at the moment is to avoid MISR data as it's more sensitive to your assumptions than the environment. I can think of three approaches to remedy this:

1. Give up and declare that your uncertainty values are uncalibrated, providing a pixel-by-pixel assessment of the relative reliability. (I'd recommend that you normalise the values to clarify that their magnitude is not inherently meaningful.)

2. Show that, despite the algorithm's theoretical sensitivity to your assumptions, the uncertainties you produce are an approximation of the true error. This would be done through a validation activity (e.g. the distribution of $(\tau_{MISR} - \tau_{AERONET})^2/\sigma_{MISR}^2$ is approximately normal).

3. Demonstrate that the sensitivity to your assumptions is small. The precise choice of types is a matter for another paper, but it's important to quantify the uncertainty's sensitivity to it. A straightforward way to do so would be re-running the retrieval with a few types removed at random.

- Sec. 4 argues that this method is good because it excludes high AOD retrievals. Could you provide some evidence that, for the two months of data you've considered, there were no large aerosol events?

P3L17 The spread of the MISR ensemble is providing a quantitative insight into the uncertainty in each retrieval *due to the assumptions made*. While the description of ensemble techniques at L9 is technically correct, ensemble techniques are used to estimate uncertainties that can't be accurately or efficiently calculated by other means. It's exceedingly rare to perturb more than one of the input data, auxiliary parameters, and underlying assumptions. Numerical weather prediction perturbs its input data in order to estimate the sensitivity of a chaotic system. Climate models perturb the auxiliary parameters because they are unknown. MISR perturbs the assumed aerosol type because the radiances available don't fully constrain the problem. MISR doesn't need to perturb the input data as the physics

of remote sensing are sufficiently linear that error propagation does a reasonable job of estimating the uncertainty due to measurement error. Hence, I wouldn't agree that extending ensembles to 'all possible sources of error' would be overly useful. Ensemble techniques are used to quantify uncertainties due to poorly understood, poorly constrained, or exceedingly non-linear error sources.

P8L38  Within this paper, the only evidence that the cloud filtering is effective is showing that mean AOD is lower. MISR is on the same platform as a MODIS, so you have the ability to check if your cloud flagging spatially agrees with them. That would be rather more convincing than the distribution of a month's observations presented in Fig. 5.

Fig.3  (b) is rather concerning. Do the peaks in retrieval count correspond to the divisions of your LUT? Also, could 3(a) and (c) be shown as 2-D histograms with the mean overplotted? Your argument would be stronger if the decrease in mean AOD with increasing ARCI is due to a decreased prevalence of large AOD (the cloud-contaminated retrievals) while the variation with $\chi^2$ is more uniform.

P9L2  This paragraph ascribes the variations in Fig. 3 at low $\chi^2$ or ARCI to poor sampling. That implies that there should be retrievals there but you didn't see them. Very low $\chi^2$ implies a very close fit to observations, which is unlikely, and very low ARCI implies a very unlikely fit, which should happen infrequently if the ensemble of aerosol types was well-chosen. Hence, I'd ascribe the sharp variations in Fig. 3 in those regions to scenes that are poorly suited to this retrieval.

P9L19  I wouldn't say that the trend in AOD is statistically robust. I'd say that the shape of 3(c) isn't evident in 3(d), so we don't ascribe the kink in the former to a change in frequency.

Fig.4  This is a superb figure and deserves more attention than Fig. 3. However, the caption is unclear if it is plotting the same data as in Fig. 3.

P10L12 Any idea why cloud contamination is a function of latitude? Does the ARCI threshold need to vary with latitude?

P12L19 You didn't provide a 'strong statistical foundation'. You justified the ARCI threshold by the shape of the distribution of AOD. Statistics would calculate a theoretically sensible value of ARCI based on typical noise and a very large ensemble of aerosol types.

- Finally, I would prefer it if the paper and any data files released clearly describe the retrieved product as 'ensemble mean AOD'. Evaluating a range of aerosol types is an excellent way to sample the unconstrained parts of state space (such as refractive index). Providing an ensemble of results to the user illustrates what the data constrains and what it doesn't. However, a combination of ensemble members doesn't necessarily have a physical meaning. To use an example from a related problem, a thick but high cloud can produce the same TOA thermal radiance as a thin and low one. Giving the user both results shows that both are possible. An ensemble mean, though, gives a medium-thickness layer midway through the atmosphere, which is inconsistent with the data.

A few more minor points:

P4L6 Perhaps 'The previous MISR dark water algorithm' would be a more informative title to someone skimming the paper?

P5L2 reflectance is  defined as

P5L26 Considering you don't define them, and their precise definition is unimportant to this paper, perhaps remove specific references to the now neglected $\chi^2$, parameters?

P6L34 'turns out to be' is rather colloquial. Perhaps 'and will be shown to produce superior results to the original algorithm'?

P7L4 If these are continuous functions of $\tau$, you are presumably interpolating as the LUT is discrete. What are you interpolating — $\rho, \chi^2$, or $f$?

P7L7 That distribution has a long tail on it to be called Gaussian. (I know what you're getting at, but there are distributions which are 'Gaussian but with a long tail'.)

Fig.2&5 Why does the scale go to 3? It washes out the global distribution of AOD while emphasising variations in retrievals that are implied to be wrong.

Fig.5 The degree symbol is underlined.

[Figure]

---

## Referee Comment (RC3) · Anonymous Referee #2 · 26 Sep 2017

Review of "New approach to the retrieval of AOD and its uncertainty from MISR observations over dark water" by M. Witek et al. for AMT

Synopsis: This paper describes a new method for retrieving AOD over water, using MISR observations. Specifically, instead of picking a retrieval solution based on the minimum cost ("best") fitting of lookup table versus observations, the new algorithm retrieves based on weighting the cost of each ensemble member. Instead of thresholds, the new retrieval is more dynamic, and appears to provide more accurate and more consistent results. Additionally, a new confidence index (known as ARCI) is proposed, which can help to screen the results. In this way, the uncertainty of the retrieval is

quantified.

Assessment: This is a good paper, and should be published after minor/medium re-
vision. The most obvious issue is that there is neither "validation" (comparison with
ground-truth, e.g. AERONET) nor detailed comparisons with other datasets (e.g.
MODIS on the same Terra platform). Based on my own experiences, I agree that
the new results seem better (lower average AOD; fewer blunders, etc). However, a
more skeptical reviewer needs some more proof including validation. I also wonder
why the previous ($\leq$ V22) retrievals had such a complicated chi-squared decision tree,
when in fact it seems to be much simpler? The paper appears to be primarily about
the advantages of the new ARCI/chi-sq metrics, which is fine. The issue becomes con-
fused when discussing new aerosol model/mixtures, and much more confused when
discussing 17.6 vs 4.4 km resolution. I recommend ONLY concentrating on the new
fitting metrics here, because that is useful enough.

Also, with the subject being the new ARCI/chi-sq metrics, I would be completely curious
to see what these look like on the globe? (function of season, perhaps?)

Writing: While the English writing is easy to read, there are issues of paragraph for-
matting (hanging vs indents). References are hard to read etc.

Specifics:

*P1L15: Why only allow AOD < 3.0? sometimes even higher?

*P2L22: Suggest using the term "confidence" rather than "quality", as the MODIS re-
trieval can't measure quality until performing validation. Confidence refers to how well
the algorithm marched through its logic steps (enough pixels? Good enough fitting?
Etc).

*P2L29: Suggest adding where these uncertainties would be useful, especially in ap-
plications of data assimilation/forecasting etc.

*P2L35: Note that the MODIS retrieval (and I think others) do not validate in terms of

±MAX(a, b x AOD), but rather as ±(a+b x AOD).

*P3L8: Ensemble approach. YES! We have more computer power, I agree! Note that the MODIS over-ocean retrieval does a poor-man's ensemble.

*P4L12: Are you reviewing the old algorithm (v22) or the new one (V23)? Or is everything common to both?

*P4L15-17: This sentence is a run-on and confusing

*P4L17: Not sure what the sentence about 1-D RT means.

*P4L36: So this more comphrensive model set is not used for V23, correct?

*P6L3: What happens to fitting error if AOD is near zero? Very low signal.

*P6L28: This sentence is a run-on.

*P6L31: What is a "blunder"? Is this a retrieval by mistake? No retrieval when should be? One with a big error? Do you really want to screen all "outliers"?

*P7L3: Does Fig. 1 represent a particular date/time/case? I know it is discussed further in a future section, but it's confusing here. At least mention that it will be discussed more. I however, like the visualization. What happens in case of bigger (or smaller) AOD? Will the spreads be smaller or larger?

*P8 last paragraph: I am getting confused because paper is discussing TWO upgrades. A. The ARCI/chi-sq stuff, and also the B. Spatial resolution (17.6 to 4.4 km). I think you need to concentrate on only A.

*P9L16: Why is low ARCI related to cloud contamination? It is definitely one reason. Could there be confusion between small ice particles and dust particles, and somehow derive a large ARCI?

*P9L20 (and Fig 3). Hard to see, because panels (b) and (d) have different y-axis scales and they are not in terms of %. To me, it looks as if there are much fewer

retrievals in panel (d) versus (b). Also, why the wiggles in (b)?

*P9L32: Is there a chance you are throwing out "good" aerosol data? Maybe you can show some AOD imagery (on a map) over-plotted on the suspected clouds?

*P9L35: Are data in Fig 4 the same data as plotted in Figs 2 and 3?

*P10L10: These are HUGE differences? Can you compare with anything (e.g. MODIS, AERONET, a model?) to prove this is reasonable? Fig 5 is nice. The "blunders" in the high latitudes (primarily around Antarctica are still glaring.

*P10L37: Fig 6. See comment from P2L35: Definitely looks like an a+bxAOD rather than MAX(a,b x AOD).

*P11L12: I am not sure that V23 uncertainties look like V22 uncertainties is useful and or a desired result.

Figures:

Fig. 3: Needs consistent y-axes between pairs of plots

Fig. 7: I am not sure this is a useful figure.

---

## Author Comment (AC2) · 11 Nov 2017

Review of "New approach to the retrieval of AOD and its uncertainty from MISR observations over dark water" by M. Witek et al. for AMT

Synopsis: This paper describes a new method for retrieving AOD over water, using MISR observations. Specifically, instead of picking a retrieval solution based on the minimum cost ("best") fitting of lookup table versus observations, the new algorithm retrieves based on weighting the cost of each ensemble member. Instead of thresholds, the new retrieval is more dynamic, and appears to provide more accurate and more consistent results. Additionally, a new confidence index (known as ARCI) is proposed, which can help to screen the results. In this way, the uncertainty of the retrieval is quantified.

Assessment: This is a good paper, and should be published after minor/medium revision. The most obvious issue is that there is neither "validation" (comparison with ground-truth, e.g. AERONET) nor detailed comparisons with other datasets (e.g. MODIS on the same Terra platform). Based on my own experiences, I agree that the new results seem better (lower average AOD; fewer blunders, etc). However, a more skeptical reviewer needs some more proof including validation. I also wonder why the previous ($\leq$ V22) retrievals had such a complicated chi-squared decision tree, when in fact it seems to be much simpler? The paper appears to be primarily about the advantages of the new ARCI/chi-sq metrics, which is fine. The issue becomes confused when discussing new aerosol model/mixtures, and much more confused when discussing 17.6 vs 4.4 km resolution. I recommend ONLY concentrating on the new fitting metrics here, because that is useful enough.

Re: We carefully considered including some form of external validation of the new approach (AODs and their pixel-level uncertainties) in this manuscript, but eventually decided the topic is challenging enough to deserve a separate study. Here we will try to briefly summarize our reasoning behind this decision. First, at the time of writing, only two months of V23 data were available, which did not provide enough comparison points against surface-based AERONET observations. At present, we have processed two years, 2014 and 2015, and obtained around 1300 collocations with AERONET. Note that we are constrained to Dark Water retrievals only, which limits the number of available AERONET locations. This number could be sufficient for AOD validation, but in our opinion it is still insufficient for a proper assessment of the reported pixel-level uncertainties. There is a range of topics that we would like to explore while assessing the MISR AOD uncertainty predictions:

- How do the spatial and temporal differences between MISR retrieval and AERONET observation influence agreement metrics?
- Is spatial variability in AOD uncertainty consistent with expectations?
- Is the AOD uncertainty dependent on specific retrieval parameters (e.g., viewing geometry, number of cameras used, ARCI parameter)?
- Is the AOD uncertainty affected by the proximity of clouds?
- How can we use information from other instruments (MODIS) to evaluate the AOD uncertainties?

These are just a few questions that we have already started investigating. In our view, a cursory evaluation within the scope of the present manuscript would have been unsatisfactory.

In this study we introduce the ARCI metric as a screening parameter and highlight its efficacy, but, in our view, this work is primarily about a new way of determining AODs and AOD uncertainties using the full information content available from the goodness-of-fit metrics. In

particular, this leads to a more plausible prediction of the AOD retrieval uncertainty, which we hope may prove useful in many aerosol modeling applications. An unwelcome side effect that we discovered after introducing this new approach was a relatively large number of high-AOD retrievals in areas that typically have low aerosol content, but at the same time are very cloudy. We concluded that these high-AOD retrievals were likely cloud contaminated due to imperfect cloud identification procedures in the MISR aerosol retrieval algorithm processing. In the V22 product, various thresholds on $\chi^2$ metrics were able to eliminate many such erroneous AOD retrievals. In V23, the new ARCI metric is a useful alternative to the V22 thresholds. The transition to a finer horizontal resolution, from 17.6 km$^2$ to 4.4 km$^2$, fundamentally increases the number of cloud-contaminated retrievals because the retrievals are often performed closer to cloud edges, and some of the cloud screening that was effective at the coarser resolution was found to be ineffective at the finer resolution. We do not, however, discuss in this manuscript the impact of the finer resolution on the quality of retrieved AODs and AOD uncertainties. This will be a subject of a separate investigation.

Also, with the subject being the new ARCI/chi-sq metrics, I would be completely curious to see what these look like on the globe? (function of season, perhaps?)
Re: Yes, this is an interesting question that we will investigate in the near future. The paper's main focus is on the new methodology for deriving AODs and AOD uncertainties in the new V23 MISR aerosol product. Including additional analysis of ARCI would, in our view, diverge the manuscript from its main topic.

Writing: While the English writing is easy to read, there are issues of paragraph formatting (hanging vs indents). References are hard to read etc.
Re: We will format the references to be more transparent.

Specifics:
*P1L15: Why only allow AOD < 3.0? sometimes even higher?
Re: MISR aerosol look up table (LUT) only includes mid-visible AODs below or equal to 3.0. It is possible to extend this range to higher AODs, but to do so requires a significant change to the LUT and adversely impacts the processing time.

*P2L22: Suggest using the term "confidence" rather than "quality", as the MODIS retrieval can't measure quality until performing validation. Confidence refers to how well the algorithm marched through its logic steps (enough pixels? Good enough fitting? Etc).
Re: Yes, we generally agree with this statement, but in this case we refer to the Quality Assurance (QA) metric specified in the MODIS product. In Levy et al. (2013) on page 2990 we read: "However, the are major changes to how data "confidence" or Quality Assurance (QA) is assigned (Hubanks, 2012)." As we refer to a flag in our sentence, we think the phrase "retrieval quality assurance flags" is appropriate.

*P2L29: Suggest adding where these uncertainties would be useful, especially in applications of data assimilation/forecasting etc.
Re: We modified the last sentence in this paragraph to read:
"While such metrics are very valuable, they comprise only crude proxies for pixel-level uncertainties and, therefore, have limited quantitative utility in applications such as aerosol forecasting and data assimilation."

*P2L35: Note that the MODIS retrieval (and I think others) do not validate in terms of ±MAX(a, b x AOD), but rather as ±(a+b x AOD).
Re: Yes, most satellite instruments retrieving AODs report their error envelopes as ±(a+b×AOD).

MISR defines the error envelop in a slightly different manner. We changed the sentence to read: "Taking the general form of ±(a+b×AOD) (or max[±a, ±(b×AOD)]), where a and b are empirically determined constants...".

*P3L8: Ensemble approach. YES! We have more computer power, I agree! Note that the MODIS over-ocean retrieval does a poor-man's ensemble.
Re: Agreed.

*P4L12: Are you reviewing the old algorithm (v22) or the new one (V23)? Or is everything common to both?
Re: We modified the sentence to read:
"Here some key elements of the V22 algorithm relevant to the new approach are reviewed.

*P4L15-17: This sentence is a run-on and confusing
Re: We rearranged this sentence to read:
"The problem of retrieving aerosol properties over large water bodies, such as oceans, seas, or deep lakes, is greatly simplified by the fact that reflectance from such surfaces is uniform and that such deep-water bodies are essentially black at red and near-infrared (NIR) wavelengths."

*P4L17: Not sure what the sentence about 1-D RT means.
Re: It is a general statement regarding the physical principle of AOD retrieval over dark water.

*P4L36: So this more comprhensive model set is not used for V23, correct?
Re: Correct, V23 includes the same set of mixtures (and same LUT) as V22.

*P6L3: What happens to fitting error if AOD is near zero? Very low signal.
Re: In Eq. 2 for $\chi^2$, the signal difference ($\rho_{MISR}-\rho_{model}$) is divided by $\sigma^2_{abs}$, defined in the text (P5L16), which takes into account the signal magnitude.

*P6L28: This sentence is a run-on.
Re: Agreed. We rearranged this sentence to read:
"The empirical thresholds in goodness-of-fit parameters in the V22 MISR dark water aerosol retrieval algorithm are used to select successful aerosol mixtures. This affects the frequency of retrieval success as well as the resulting AODs, AOD uncertainties, and aerosol properties."

*P6L31: What is a "blunder"? Is this a retrieval by mistake? No retrieval when should be? One with a big error? Do you really want to screen all "outliers"?
Re: A retrieval "blunder" is a retrieval with very high AOD that is untrustworthy and possibly affected by cloud contamination. Reasons other than cloud contamination are also possible. Ideally, cloud identification procedures should be able to eliminate all cloud-contaminated pixels so that an aerosol retrieval is not performed. However, most satellite instruments suffer to some extent from erroneous cloud identification, in which case cloudy pixels are used in aerosol retrievals. This results in clouds being retrieved as aerosols with unreasonably high AODs.

*P7L3: Does Fig. 1 represent a particular date/time/case? I know it is discussed further in a future section, but it's confusing here. At least mention that it will be discussed more. I however, like the visualization. What happens in case of bigger (or smaller) AOD? Will the spreads be smaller or larger?
Re: This is a randomly selected case. We added appropriate clarification in the text:

"The key elements of the new method are visualized in Figure 1 using actual MISR data from a randomly selected case."
Generally yes, the spread, and the uncertainty, depends on the retrieved AOD, which we visualize in Fig. 6.

*P8 last paragraph: I am getting confused because paper is discussing TWO upgrades. (A) The ARCI/chi-sq stuff, and also the (B) Spatial resolution (17.6 to 4.4 km). I think you need to concentrate on only (A).
Re: The increased resolution of the retrieval is not an upgrade that we are concentrating on in this manuscript. The processing pathway is exactly the same in both the 17.6 and 4.4 km retrievals, except that the 4.4 km retrieval covers a smaller area. In fact, the MISR Dark Water algorithm at either resolution selects only one 1.1 km pixel, which is then used to perform an aerosol retrieval. This one pixel in V22 is assumed to represent an area of 17.6 x 17.6 km, whereas in V23 it represents an area that is 16 times smaller (4.4 x 4.4 km). This is why retrievals are often performed closer to cloud edges.

*P9L16: Why is low ARCI related to cloud contamination? It is definitely one reason. Could there be confusion between small ice particles and dust particles, and somehow derive a large ARCI?
Re: In our analysis we observed a relationship between the prevalence of high-AOD retrievals and low ARCI. These high-AOD retrievals are in areas that climatologically have very low aerosol burdens, but are characterized by high cloud coverage. Cloud contamination in the MISR retrieval appears to be the most plausible explanation for such high-AOD results.
There are certain conditions when the MISR retrieval algorithm identifies thin cirrus clouds as non-spherical mineral dust mixtures. This was documented in a study by Kalashnikova et al. (2013) and manifests itself as bands of aerosol nonsphericity over high latitude oceans (e.g., the Southern Ocean, Northern Atlantic) that shift with the seasons. This is clearly an issue of cloud contamination. Those retrievals, however, tend to have low ARCI, and the new screening approach based on the ARCI threshold is able to eliminate them.

*P9L20 (and Fig 3). Hard to see, because panels (b) and (d) have different y-axis scales and they are not in terms of %. To me, it looks as if there are much fewer retrievals in panel (d) versus (b). Also, why the wiggles in (b)?
Re: The maximum values in Fig. 3(b) and (d) are different, but the scale is linear in both cases. We concentrate on the trend in retrieval count, rather than on the absolute values, which depend on the spacing of the ARCI and $min(\chi^2)$ parameters. In this particular case, we used 200 intervals for $min(\chi^2)$ (range from 0 to 5), and 290 intervals for ARCI (range from 0.013 to 0.4).
We do see certain clustering around specific $min(\chi^2)$ values in our dataset, which gives rise to small wiggles seen in Fig. 3b. This is probably related to the finite AOD gridding of our LUT, which is 0.025 throughout most of the AOD range. We plan to investigate this feature in greater detail in the future. Furthermore, the wiggles in Fig. 3b become apparent only because of very fine sampling of the $min(\chi^2)$ space, which is 0.025 in this case.

*P9L32: Is there a chance you are throwing out "good" aerosol data? Maybe you can show some AOD imagery (on a map) over-plotted on the suspected clouds?
Re: We have not looked at particular cases or extensively investigated specific regimes in Fig. 4a. However, motivated by your comment we looked at the origin of this particular group of high-AOD retrievals with $min(\chi^2)$ around 0.2 and ARCI around 0.1. This turns out to be about ~410 retrievals coming from one orbit in 18[th] January 2007 (orbit 37689). To our surprise, these retrievals are south of the Ivory Coast, Africa. The figure below shows unscreened AODs from MISR V23. There are some scattered clouds in the scene but they are not related to the patches of

high-AOD (>2.0) retrievals. The aerosol background is high with AODs exceeding 0.5. The second figure shows MISR equivalent reflectances from the red wavelength for the same scene. This is to show that the "plume" of high-AOD in the first figure does not correspond to the higher radiances measured by the instrument. The visible imagery from MODIS also corroborates the finding that there is no substantially thicker aerosol plume is this area. This strongly suggests that the retrieved AODs in this region are retrieval artifacts, likely related to the mismatch in assumed aerosol properties between the current MISR LUT and reality, which may be a smoke and dust mixture not contained in the current MISR LUT. The current ARCI threshold screens out these "poor" retrievals.

[Figure]

**Figure 1 MISR V23 unscreened AOD from orbit 37689, blocks 87-88, time: January 18, 2007, 10:58 UTC. The high-AOD retrievals in the center right of the image have low ARCI and are therefore screened out in the final product. These retrievals, however, have min($\chi^2_{abs}$) values below 2.0 and therefore would have passed the in the previous V22 algorithm.**

[Figure]

**Figure 2 MISR red band equivalent reflectances for the same scene as in Fig. 1. Radiance data does not support the very high-AOD plume indicated by V23 aerosol retrievals.**

[Figure]

Figure 3 MODIS visible composite for the similar scene as in Fig. 1, with the red oval highlighting an approximate location of the high-AOD and low ARCI retrievals in MISR V23. A substantially thicker aerosol plume is not visible in the MODIS imagery.

*P9L35: Are data in Fig 4 the same data as plotted in Figs 2 and 3?
Re: Yes. We clarified it in the text and in the caption to Fig. 4.
"Another way to look at the difference between the two screening approaches is presented in Fig. 4a, which shows the two-dimensional distribution of average AOD as a function of min( $\chi^2_{abs}$ ) and ARCI using combined data from January and July of 2007."

"Figure 4 (a) average AOD as a function of ARCI and min( $\chi^2_{abs}$ ) for the combined months of January and July of 2007…"

*P10L10: These are HUGE differences? Can you compare with anything (e.g. MODIS, AERONET, a model?) to prove this is reasonable? Fig 5 is nice. The "blunders" in the high latitudes (primarily around Antarctica are still glaring.

Re: The difference in global average AOD is large indeed, but the value for the unscreened data is clearly unrealistic. This indicates the impact of ARCI screening on the product. MODIS would give a somehow similar number to the screened V23 product (~0.14).
The speckles of high-AOD in some remote areas are still present but they are addressed by an additional screening procedure not discussed in this paper.

*P10L37: Fig 6. See comment from P2L35: Definitely looks like an a+bxAOD rather than MAX(a,b x AOD).
Re: Yes, this appears to be the case here. We will establish the relationship in the upcoming external validation work.

*P11L12: I am not sure that V23 uncertainties look like V22 uncertainties is useful and or a desired result.
Re: Since AOD uncertainties are reported in the previous V22 MISR aerosol product, it was natural to compare the new V23 uncertainty estimates to the previous ones.

Figures:
Fig. 3: Needs consistent y-axes between pairs of plots
Re: See our response to comment P9L20.

Fig. 7: I am not sure this is a useful figure.
Re: AOD uncertainties have been reported in all versions of the MISR aerosol product. Some readers who have previously looked at this parameter might find it instructive to compare the new predictions with those from V22.

---

## Author Comment (AC3) · 11 Nov 2017

This manuscript details a new method to retrieve AOD and pixel-level AOD uncertainty from the Multi-angle Imaging SpectroRadiometer. The manuscript is well-written, and it is likely that this new method will serve the MISR aerosol community well in the future. However, I can not support publication of this work without at least a cursory attempt at validation, which this paper is sorely missing.

Major Comment: Although I believe that this new method probably represents a substantial improvement for aerosol remote sensing from MISR, it is incumbent on the authors to prove this. The authors claim that not all sources of error are included, therefore no comparison of the AOD differences and pixel-level uncertainties (against MAN or AERONET, for instance) should be done. If this new method is going to be implemented in the next version of the MISR aerosol retrieval algorithm (and going to be published here), it should be validated first. A comparison of new algorithm results with old algorithm results does not replace real validation. Re: We carefully considered including some form of external validation of the new approach (AODs and their pixel-level uncertainties) in this manuscript, but eventually decided the topic is challenging enough to deserve a separate study. Here we will try to briefly summarize our reasoning behind this decision. First, at the time of writing, only two months of V23 data were available, which did not provide enough comparison points against surface-based AERONET observations. At present, we have processed two years, 2014 and 2015, and obtained around 1300 collocations with AERONET. Note that we are constrained to Dark Water retrievals only, which limits the number of available AERONET locations. This number could be sufficient for AOD validation, but in our opinion it is still insufficient for a proper assessment of the reported pixellevel uncertainties. There is a range of topics that we would like to explore while assessing the MISR AOD uncertainty predictions:

- How do the spatial and temporal differences between MISR retrieval and AERONET observation influence agreement metrics?
- Is spatial variability in AOD uncertainty consistent with expectations?
- Is the AOD uncertainty dependent on specific retrieval parameters (e.g., viewing geometry, number of cameras used, ARCI parameter)?
- Is the AOD uncertainty affected by the proximity of clouds?
- How can we use information from other instruments (MODIS) to evaluate the AOD uncertainties?

These are just a few questions that we have already started investigating. In our view, a cursory evaluation within the scope of the present manuscript would not have been unsatisfactory.

**Minor Comment:**

The authors mention (5 times) that several of the thresholds from the current version of the MISR aerosol retrieval algorithm are arbitrary. Please refrain from so much hyperbole in the manuscript, as most empirical thresholds could be considered arbitrary (including your own ARCI threshold). Re: We eliminated most occurrences of the phase "arbitrary thresholds" from the manuscript and substituted it with "empirical thresholds". The one remaining case is on page 12, line 14: "This approach allows for a consistent calculation of AOD and AOD uncertainty without the need for screening acceptable mixture solutions based on a complex interplay of multiple, and somewhat arbitrary, thresholds."

---

## Author Comment (AC1)

This paper describes a change to the MISR aerosol retrieval algorithm. They select an ensemble of aerosol types and, for each, compute the radiances that would be observed at a range of aerosol optical depths (AOD). Previously, ensemble members were evaluated separately so each gave an AOD and cost, which were then filtered and averaged to calculate the final product. This paper proposes minimising a single cost function (being the sum of the individual cost functions) to find the AOD and it's uncertainty. The technique is rationalised based on two months of observations and is shown to produce more believable uncertainties, on average, than the previous algorithm.

I recommend this paper for publication after minor revisions. The technique proposed is definitely a step in the right direction and the paper is superbly drafted. However, the technique and description thereof could be improved by a more statistical approach. The paper justifies itself with qualitative descriptions of global averages and internal metrics rather than any validation activity, which is common but always disappointing. Specific comments on the paper are listed below, with some minor details collected at the end. The notation PxLy refers to line y of page x.

My experience is in optimisation. One defines a cost function and selects an algorithm to efficiently search the 'surface' of that function for its global minima. The uncertainty is a measure of the 'width' of that minima in multi-dimensional space (i.e. the magnitude by which a variable could be changed without significantly increasing the cost). The cost function is usually the RMS difference between some modelled value and a measurement. If the model is accurate and the measurement suffers only random noise (of known variance), the minimal value of the cost function will sample a χ2 distribution, from which one can determine the probability that this measurement fit that model. To me, this paper essentially proposes that *f*(τ) is a probability density function (PDF) for AOD and that it is normally distributed. It follows that the most likely AOD is the τ that maximises *f* and the uncertainty is the function's width. The proposed ARCI threshold can then be understood as eliminating retrievals that are exceedingly unlikely. Describing the problem with these basic statistical concepts could vastly simplify the paper, avoiding awkward phrasing like P8L5.

Re: It is a very valuable observation. We added the following clarification below Eq. 4. "The function f can be interpreted as a probability density function (PDF) for AOD. The most likely AOD is the one that maximizes f (Eq. 4), and the retrieval uncertainty is related to the width of the PDF."

We also modified the somehow awkward phrasing in P8L5 to read: "Large ARCI, on the other hand, means that for some models sufficiently low  $\chi^2_{abs}$  were obtained, signifying good agreement with the observations."

• Because this is a fairly straightforward statistical problem, there exists a variety of tools to check that (a) f is in fact a good model of the PDF, (b) f is normally distributed, and (c) the selected aerosol models are an unbiased sampling of the complete state space of real-world aerosols. A brief discussion of some of those points could provide a standardised means to evaluate your assumptions and avoid qualitative judgements, such as the function 'closely resembles a Gaussian' (P7L12).

Re: In the process of designing and testing the new approach, at one point we did fit a normal distribution to our PDF results. We compared most likely AODs retrieved from PDFs against those retrieved from the fitted normal distributions. The results were in

excellent agreement. This exercise gave us confidence that, at least in those cases that we considered, the PDFs closely resembled Gaussian distributions. However, we though this analysis was too technical to be included in the manuscript. As for point (c), we write in the manuscript that the resulting uncertainty is dependent on the LUT considered in the retrieval (P10L30-35) and that the 74 mixtures currently included in MISR retrieval process are not complete (P4L35-38).

- Are you tabulating f as a function of linear or log τ? Figure 1 uses both as an x-axis, which is misleading. It should be logarithmic as AOD is log-normally distributed (which is clear from the asymmetry about τmax in Fig.1(3)). If you're using linear space, you will underestimate the uncertainty and overestimate the mean.
   Re: All equations in the manuscript use linear τ. In Figure 1a we use the logarithmic scale in the x-axis to better visualize the cost functions at very low τ. Because after inverting the cost functions, at low τ the signal becomes very small, the log scale is no longer necessary. We added additional clarification regarding the x-axis scale in the caption. The distribution in Fig. 1c is close to Gaussian. The misleading resemblance to a log-normal distribution comes from the fact that the PDF is truncated on the left side due to the physical constraint (τ>0.0).
- Why is there no validation of the new algorithm? It seems fairly substantial to move from averaging a few aerosol types per pixel to averaging 74. A few comparisons against AERONET or MODIS would be fine for a paper like this. A simple comparison of V22 vs. V23 would be a start, considering you did it for the uncertainty!
   Re: A validation paper is currently under preparation. It was our intention to designate external validation efforts to a separate publication. One reason for this is that, at the time of preparing this manuscript, we only had two months of data available, which is not enough to obtain sufficient number of collocations with ground based observations. Furthermore, we plan to investigate the new AODs and their pixel-level uncertainties in greater detail, which we feel justifies a separate study. Our analysis indicates that the new algorithm leads to AODs that are similar, but not identical, to those obtained using thresholds from V22. However, the uncertainty quantification in the new approach is sufficiently different from V22 to justify a comparison figure (Fig. 7).
- In Sec. 3, you implicitly assume that the choice of aerosol type overwhelms any measurement error. Could Fig. 1 be adapted to show the sensitivity of a  $\chi^2$  curve to typical measurement error? I'd expect it to move the curve slightly, but much less than the spread between curves.

Re: The measurement error is embedded in the calculation of  $\chi^2_{abs}$  (Eq. 2). The absolute radiometric uncertainty  $\sigma_{abs}$  in V22 is set to 5% of the signal itself for each camera and wavelength (P5L16). We feel that showing the sensitivity of  $\chi^2_{abs}$  to different levels of  $\sigma_{abs}$  would decrease the clarity of the figure.

P10L24 I'm unhappy with this paragraph.

• L27 I think this is trying to distinguish between a validation activity, which you sadly aren't doing, and an uncertainty estimate, which you are. By definition, uncertainty is a parameter describing the range of values that can be reasonably ascribed to the quantity that is being measured. I believe that provides a 'measure of how far the retrieved AOD deviates from the "truth". The distinction is that uncertainty is a prediction of that difference while validation is a direct calculation of it.

Re: What we are trying to distinguish here is the algorithmic retrieval uncertainty on the one hand, and the uncertainty that comes from comparing a retrieved AOD with ground truth on the other hand. In both cases we are considering pixel-level information, or individual retrievals, rather than a bulk validation metric like the error envelope. Yes, we are predicting an uncertainty in our algorithm, but this prediction might not necessarily represent the real range of values that are being measured. We are trying to be cautious here and not assign undue credit and value to the algorithm's prediction. A validation activity is required to establish the relationship between the reported uncertainties and the ground truth. Because we think this is a challenging task, we left it for a separate investigation. Our initial results, however, show very promising linkage between our reported uncertainty and the standard deviation of a normally distributed error function.

- It's good to be clear that the estimated uncertainty is sensitive to the way you solve the problem. However, you don't tell the user what to do with that information. I think a rational response at the moment is to avoid MISR data as it's more sensitive to your assumptions than the environment. I can think of three approaches to remedy this:
  - 1. Give up and declare that your uncertainty values are uncalibrated, providing a pixelby-pixel assessment of the relative reliability. (I'd recommend that you normalise the values to clarify that their magnitude is not inherently meaningful.)
  - 2. Show that, despite the algorithm's theoretical sensitivity to your assumptions, the uncertainties you produce are an approximation of the true error. This would be done through a validation activity (e.g. the distribution of  $(\tau_{MISR} \tau_{AERONET})^2 = \sigma^2_{MISR}$  is approximately normal).
  - 3. Demonstrate that the sensitivity to your assumptions is small. The precise choice of types is a matter for another paper, but it's important to quantify the uncertainty's sensitivity to it. A straightforward way to do so would be re-running the retrieval with a few types removed at random.

Re: Indeed, at the moment our uncertainty values are uncalibrated. But this is a temporary position that will be resolved in a separate investigation. In order to validate our uncertainties, large comparison statistics against ground truth are required. As mentioned above, at the time of writing we did not have enough data (two months of retrievals) and enough collocations against AERONET to perform a detailed evaluation of the retrieved parameters. This activity will be performed along with the reprocessing of the MISR mission with the new V23 version of the aerosol product.

• Sec. 4 argues that this method is good because it excludes high AOD retrievals. Could you provide some evidence that, for the two months of data you've considered, there were no large aerosol events?

Re: Figure 5 shows the global distribution of AOD with ARCI screening for January and July of 2007. There are high AOD regions visible off the west coast of Africa and off the coasts of India and China. These are associated with high-AOD events such as dust outflow from Africa, biomass burning, and anthropogenic emissions. We write in the manuscript: (P10L7) "At the same time, climatologically large AODs off the coasts of Africa and South and East Asia are retained, indicating that the new screening method does not unintentionally remove all high AODs that are likely valid."

P3L17 The spread of the MISR ensemble is providing a quantitative insight into the uncertainty in each retrieval due to the assumptions made. While the description of ensemble techniques at L9 is technically correct, ensemble techniques are used to estimate uncertainties that can't be accurately or efficiently calculated by other means. It's exceedingly rare to perturb more than one of the input data, auxiliary parameters, and underlying assumptions. Numerical weather prediction perturbs its input data in order to estimate the sensitivity of a chaotic system. Climate models perturb the auxiliary parameters because they are unknown. MISR perturbs the assumed aerosol type because the radiances available don't fully constrain the problem. MISR doesn't need to perturb the input data as the physics of remote sensing are sufficiently linear that error propagation does a reasonable job of estimating the uncertainty due to measurement error. Hence, I wouldn't agree that extending ensembles to 'all possible sources of error' would be overly useful. Ensemble techniques are used to quantify uncertainties due to poorly understood, poorly constrained, or exceedingly non-linear error sources. Re: We modified this sentence to read: "Such an approach, if extended to all poorly quantifiable nonlinear sources of error and physically plausible realizations of parameter space, has the potential of providing a robust and comprehensive measure of retrieval uncertainty in the manner suggested by Povey and Grainger (2015)."

- P8L38 Within this paper, the only evidence that the cloud filtering is effective is showing that mean AOD is lower. MISR is on the same platform as a MODIS, so you have the ability to check if your cloud flagging spatially agrees with them. That would be rather more convincing than the distribution of a month's observations presented in Fig. 5. Re: Yes, it could potentially be convincing to compare our screening method against MODIS. However, comparing different cloud screening techniques between satellite instruments, even on the same platform, is quite challenging and in our opinion it would extend beyond the scope of this study. MISR and MODIS have different spectral bands with different calibrations, different spatial resolutions, and the data are projected differently. The Global Energy and Water cycle Experiment (GEWEX) has an extensive report that describes such instrumental differences and compares their cloud products available online (http://climsery.ipsl.polytechnique.fr/gewexca/). The point being made in the manuscript is that the ARCI-based retrieval screening provides first line of defense against cloud-contaminated retrievals. Additional screening steps using other types of information are applied to filter out more retrievals potentially contaminated by clouds. These cloud-screening steps will be described in a separate publication.
- Fig.3 (b) is rather concerning. Do the peaks in retrieval count correspond to the divisions of your LUT? Also, could 3(a) and (c) be shown as 2-D histograms with the mean overplotted? Your argument would be stronger if the decrease in mean AOD with increasing ARCI is due to a decreased prevalence of large AOD (the cloud-contaminated retrievals) while the variation with  $\chi^2$  is more uniform.

Re: We do see certain clustering around specific  $\min(\chi^2)$  values in our dataset, which gives rise to the small wiggles seen in Fig. 3b. This is probably related to the finite AOD gridding of our LUT, which is 0.025 throughout most of the AOD range. We plan to investigate this feature in greater detail in the future. Furthermore, the wiggles in Fig. 3b become apparent only because of very fine sampling of the min( $\chi^2$ ) space. Our interval is 0.025, which results in 200 data points for min( $\chi^2$ ) between 0 and 5.

We created a 2-D figure with results from Fig. 3c by plotting normalized histograms of AOD at each ARCI level. An example is presented below in Figure 1. The black solid and dashed lines are the mean and the median AODs. The figure does show decreasing number of high AODs with increasing ARCI, but the results are not as clearly visible as in Fig. 3 in the manuscript. Another useful metric showing that the number of cloud-contaminated high-AOD retrievals is decreasing with increasing ARCI is the percentage of retrievals with AODs higher than 2.0. This is presented below in Figure 2. The percentage of high-AOD retrievals decreases from the top 16% at ARCI=0.03 to about 1% at ARCI=0.15.

Figure 2 below is simple and clearly conveys the message, but we decided a description in the text was sufficient to strengthen our argument.

"In the first regime, the average AOD is highly sensitive to the specific value of ARCI, characterized by a sharp decrease in AOD with increasing ARCI between about 0.03 and 0.13. This suggests that a decreasing number of cloud-contaminated, high-AOD retrievals are included in the average as the ARCI is increased. Indeed, the percentage of retrievals with AOD higher than 2.0 reaches its peak, 16%, at ARCI equal to 0.03, and decreases to about 2% when ARCI is 0.13. In the second regime..."

P9L2 This paragraph ascribes the variations in Fig. 3 at low  $\chi^2$  or ARCI to poor sampling. That implies that there should be retrievals there but you didn't see them. Very low  $\chi^2$  implies a very close fit to observations, which is unlikely, and very low ARCI implies a very unlikely fit, which should happen infrequently if the ensemble of aerosol types was well-chosen. Hence, I'd ascribe the sharp variations in Fig. 3 in those regions to scenes that are poorly suited to this retrieval.

Re: We agree with this statement. We ascribed the variations at low  $\chi^2$  or ARCI to poor sampling without providing an explanation of why the sampling in these regimes is low. We did not want to put too much emphasis in our analysis to these low  $\chi^2$  or ARCI regimes, as they are not very relevant to our main arguments. We did, however, change the phrasing in this paragraph:

"After a rapid initial drop related to a similar rapid increase in sampling..." "After excluding the initial fluctuation for extremely small ARCI related to poor sampling..."

P9L19 I wouldn't say that the trend in AOD is statistically robust. I'd say that the shape of 3(c) isn't evident in 3(d), so we don't ascribe the kink in the former to a change in frequency. Re: Agreed. We changed this sentence to read: "The retrieval count decreases slowly with increasing ARCI (Fig. 3d), indicating that the observed trends in the average AOD cannot be ascribed to a change in frequency."

Fig.4 This is a superb figure and deserves more attention than Fig. 3. However, the caption is

unclear if it is plotting the same data as in Fig. 3.

Re: We clarified the data used in the caption and in the text:

"Another way to look at the difference between the two screening approaches is presented in Fig. 4a, which shows the two-dimensional distribution of average AOD as a function of min( $\chi^2_{abs}$ ) and ARCI using combined data from January and July of 2007."

"Figure 4 (a) average AOD as a function of ARCI and min( $\chi^2_{abs}$ ) for the combined months of January and July of 2007..."

P10L12 Any idea why cloud contamination is a function of latitude? Does the ARCI threshold need to vary with latitude?

Re: Global cloud fraction has a strong latitudinal component due to the patterns of global atmospheric circulation. We have not investigated possible variations of the ARCI threshold with latitude. We will consider this possibility in the future.

P12L19 You didn't provide a 'strong statistical foundation'. You justified the ARCI threshold by the shape of the distribution of AOD. Statistics would calculate a theoretically sensible value of ARCI based on typical noise and a very large ensemble of aerosol types.
Re: Agreed. We changed this sentence to read:
"Alth each this ensemble of aerosol is an an art aliminate all AOD sufficient it is superior.

"Although this screening method does not eliminate all AOD outliers, it is superior to the previously used thresholds in V22 of the MISR aerosol product."

Finally, I would prefer it if the paper and any data files released clearly describe the retrieved product as 'ensemble mean AOD'. Evaluating a range of aerosol types is an excellent way to sample the unconstrained parts of state space (such as refractive index). Providing an ensemble of results to the user illustrates what the data constrains and what it doesn't. However, a combination of ensemble members doesn't necessarily have a physical meaning. To use an example from a related problem, a thick but high cloud can produce the same TOA thermal radiance as a thin and low one. Giving the user both results shows that both are possible. An ensemble mean, though, gives a medium-thickness layer midway through the atmosphere, which is inconsistent with the data. Re: The new V23 data product labels the retrieved AOD as, simply, "Aerosol\_Optical\_Depth". To a sophisticated user, the idea that this is essentially a "ensemble mean AOD" is a useful concept, which is one of the reasons for writing this manuscript. However, as this AOD is the one the MISR project would like the majority of users to work with, we elected to eliminate the jargon and provide a simpler designation for this field.

A few more minor points:

- P4L6 Perhaps 'The previous MISR dark water algorithm' would be a more informative title to someone skimming the paper?Re: We changed the title of the second section to: "Previous MISR V22 dark water algorithm"
- P5L2 reflectance is defined as Re: Corrected.
- P5L26 Considering you don't define them, and their precise definition is unimportant to this

paper, perhaps remove specific references to the now neglected  $\chi^2$ , parameters? Re: We want to make sure that the readers are aware of additional metrics and thresholds used in V22 processing. This is important as the new approach simplifies the process considerably and makes it more transparent.

- P6L34 'turns out to be' is rather colloquial. Perhaps 'and will be shown to produce superior results to the original algorithm'?
- Re: Agreed. We modified this sentence to read: "Furthermore, it results in a single parameter that enables screening of retrieval blunders and AOD outliers and which outperforms results derived using the original V22 thresholds."
- P7L4 If these are continuous functions of  $\tau$ , you are presumably interpolating as the LUT is discrete. What are you interpolating  $\rho$ ;  $\chi^2$ ; or f? Re: We interpolate  $\chi^2$